# Genetic Diversity of *Pseudomonas syringae* pv. *actinidiae*: Seasonal and Spatial Population Dynamics

**DOI:** 10.3390/microorganisms8060931

**Published:** 2020-06-20

**Authors:** Daniela Figueira, Eva Garcia, Aitana Ares, Igor Tiago, António Veríssimo, Joana Costa

**Affiliations:** 1Department of Life Sciences, Centre for Functional Ecology, University of Coimbra, Calçada Martim de Freitas, 3000-456 Coimbra, Portugal; dfigueira@ipn.pt (D.F.); egarcia@ipn.pt (E.G.); ayebra@ipn.pt (A.A.); itiago@uc.pt (I.T.); averissimo@uc.pt (A.V.); 2FitoLab, Laboratory for Phytopathology, Instituto Pedro Nunes, 3030-199 Coimbra, Portugal

**Keywords:** *Actinidia chinensis* var. *deliciosa*, *Pseudomonas syringae* pv. *actinidiae*, population genetic structure, endophytic and epiphytic populations

## Abstract

*Pseudomonassyringae* pv. *actinidiae* (Psa) is a gram-negative bacterium responsible for the bacterial canker in *Actinidia chinensis* var. *deliciosa* and *A. chinensis* var. *chinensis*, a quarantine organism threatening the kiwifruit industry sustainability. The present study aimed to determine the genetic structure of the endophytic and epiphytic populations of Psa isolated from four different Portuguese orchards with distinct abiotic conditions in two consecutive seasons. The results identified several coexisting and highly heterogeneous Psa populations. Moreover, evident changes in population structure occurred between the epiphytic and endophytic populations, and between seasons with a notable decrease in Psa diversity in autumn. This work provided solid evidence that the initial clonal expansion of Psa in Europe was followed by a wide genomic diversification. This perspective is important for the understanding of kiwifruit bacterial canker disease occurrence and Psa evolution, namely when adopting strategies for management of epidemics.

## 1. Introduction

The bacterial canker disease of *Actinidia* spp., caused by the bacterium *Pseudomonas syringae* pv. *actinidiae* (Psa), has been affecting kiwifruit orchards worldwide for the past decade. The severity of the disease and lack of control led to a pandemic causing significant losses in production, risking the economic viability of the sector [1,2,3,4].

Psa infection was first described in 1984 in Japan [5]. In Europe, Italy was the first country to report the presence of this bacterium in 1992, where no considerable losses were reported on “Hayward” orchards [1]. However, in 2008, the first highly destructive epidemic occurred in Italy, resulting from a single introduction, devastating the susceptible cultivars of yellow kiwifruits “Hort16” and “Jin Tao”, as well as the more resistant green kiwifruit cultivar “Hayward” [1,6]. Between 2011 and 2012, the disease rapidly spread to the rest of the European and Mediterranean Plant Protection Organization (EPPO) producing countries [2]. To contain the epidemic across Europe, Psa has been included on the A2 list for plant quarantine organisms from EPPO [7].

As no curative methods are available, control strategies are based mostly on preventive and mitigating measures [8], that is, when the disease is detected, the mitigation phytosanitary management practices of the orchards aim to decrease the inoculum in an attempt to reduce losses and prevent the spread [9]. Pruning of symptomatic plants, copper-based compounds, and elicitors are commonly used, although with limited success [8].

Recent studies based on genetic and biochemical analysis confirmed the existence of at least six different Psa types (also known as biovars) capable of infecting *Actinidia* spp., but with different levels of virulence [10,11,12]. Psa biovar 1 strains were associated with the initial epidemic outbreak of the disease in Japan (1984–1989) and Italy (1992) [13] and contain in their genome the phaseolotoxin gene *argK-tox* cluster, responsible for the formation of the chlorotic halo lesions [14]. This operon is absent in strains of biovar 2, which have been only associated with South Korea disease outbreaks [14,15]. On the other hand, these strains can produce coronatine, a non-host specific phytotoxin [16]. The predominant Psa population isolated since the 2008 outbreak in Italy, and currently spread throughout Europa, belongs to the pandemic biovar 3, responsible for substantial economic losses worldwide [9,17]. This pandemic biovar derived from a Psa population in China presents a highly stable core genome and has recently undergone worldwide expansion [18]. Recent studies based on biovar 3 strains isolated in Europe suggested the existence of variant strains, with reduced virulence, that coexists with virulent strains in mixed populations [19].

A fourth biovar has been described comprising strains unable to cause systemic infections or plant death, causing only leaf spots [20]. Biovar 4 has been isolated in Australia, New Zealand, France, and more recently in Spain [10,20,21,22]. Given the pathogenic differences with the remaining biovars, Cunty and colleagues [21] proposed a new *P. syringae* pathovar comprising biovar 4 strains named *Pseudomonas syringae* pv. *actinidifoliorum*.

Recently, two new Psa populations (biovar 5 and 6) were identified in Japan [11,12]. Biovar 5 strains are phylogenetically related to biovar 2 strains but lack the coronatine biosynthetic genes. Biovar 6 strains proved to be different from the above-mentioned populations, although being closely related to strains belonging to biovar 1 and biovar 3. Strains from biovar 6 are characterized by the presence of genes coding for phaseolotoxin and coronatine and the lack of genes for the effectors *hopH3* (specific to biovar 1 strains), *hopH1* and *hopZ5* (specific to biovar 3) [12]. Multiple biovars and lineages of Psa have been isolated in Japan and Korea, suggesting they are likely the source of all Psa populations known so far [18,23].

Nowadays, the mechanisms of penetration and colonization of Psa are better understood, and beside the susceptibility of the host, are modulated by biotic and abiotic conditions [4,24,25]. Indeed, temperature and humidity contribute decisively to the severity and dispersal of the disease [4,8]. Host plant colonization can occur throughout the year; nevertheless, bacterial growth is favoured in certain seasons, while temperatures between 12 °C and 18 °C and high humidity foster Psa multiplication, growth is reduced above 25 °C and below 10 °C [26,27]. Moreover, the plant microbiome can also influence plant immunity response and/or pathogen virulence [25,28].

Psa is known to survive on the surface of several plant organs, including buds, leaves, twigs, and flowers [9,29]. For a successful infection cycle, Psa strains need to surpass host plant defences, making a transition from the epiphytic phase to the endophytic phase to cause disease [25]. Indeed, Psa can enter the host plant through natural surface structures or wounds, like stomata, trichomes, lenticels, and fruit abscission scars or fresh cuts, and endophytically migrate from the leaves to shoots and canes via apoplast [4,24,30]. Once inside the host’s tissues, Psa can migrate systemically to different organs preferentially through xylematic vessels [4]. Endophytic colonization occurs only after Psa epiphytic population exceeds the infection threshold that varies according to plant susceptibility [4]. Nevertheless, it has been reported that epiphytic Psa populations can survive over 21 weeks on the leaf surface without causing any symptom [9,29]. Moreover, different Psa biovars developed different accessorial virulence factors to surpass host defences, resulting in distinct degrees of virulence [25]. Despite the major impact of this disease, studies aiming to determine the genetic diversity of Psa on an attempt to establish epidemiological links used a limited number of strains [10,13,19,23,31,32,33,34,35]. These reports revealed limited genetic variability that may reflect the fitness variation in the ability of Psa strains to persist, penetrate, and colonize host plants. Thus, a comprehensive characterization of Psa genetic population structure is still absent.

Psa-mediated bacterial canker in Portugal was first described in 2010 on two-year-old plants of *A. deliciosa* cv. “Summer” kiwifruit orchards, in the northern region, more specifically in Santa Maria da Feira and Valença [36]. In 2011, Psa was detected in plants imported from Italy and, during that year, new disease outbreaks were reported in other orchards from the same Portuguese region [37]. Nowadays, the disease continues to spread through the northern and central regions. Despite the limited characterization of the Psa populations in Portugal, some studies revealed that the Portuguese Psa isolates belonged to biovar 3, but some degree of genetic variability has been determined between the few tested isolates [37,38,39].

The present study aimed to determine the genetic structure of the endophytic and epiphytic populations of Psa from a culture collection of six-hundred and four strains isolated from four different Portuguese orchards characterized by distinct abiotic conditions in two consecutive seasons. In this study, several coexisting and highly heterogeneous Psa populations were identified. Moreover, changes in the population structure between seasons and between the epiphytic and endophytic Psa populations were evident. This work provides solid evidence that the initial clonal expansion of Psa in Europe was followed by a wide genomic diversification probably related to selective pressures like differences in the environmental conditions, epidemic dynamics, and disease management strategies. Future comparative genomics studies and pathogenicity tests will provide insight into how this variability translates into the cause and spread of the disease. These data are crucial to assess the risk associated with infection in each orchard and to adapt cultural practices to its correct mitigation.

## 2. Materials and Methods

### 2.1. Field Surveys

Four orchards of *Actinidia deliciosa* from different areas of continental Portugal were selected for the presence of *Pseudomonas syringae* pv. *actinidiae* (Psa) based on the region, age, degree of Psa severity, and cultivar (Table 1). In detail, two orchards were in the northern region of Portugal—Entre Douro e Minho region, orchard A in Viana do Castelo and orchard B in Guimarães. Two other orchards were located in Coimbra, in the central region of Portugal, orchard C and orchard D (Appendix A). All orchards were *Actinidia chinensis cv. deliciosa* cultivar ‘Hayward’, except for orchard A, which was *A. chinensis cv. deliciosa* cultivar ‘Bo Erica’. The incidence of the disease was evaluated based on a severity degree scale (adapted from Cunty et al. [21]), which was attributed to each kiwifruit orchard according to the observed symptoms. The incidence varied between orchards, with orchard C being the most affected, followed by orchard B and D, while orchard A was the least affected, presenting only mild infection. Moreover, orchard A was selected for the study because it was where Psa was first detected in 2010 in Portugal. The same sampling procedure was carried out on the same tagged kiwifruit plants on two consecutive occasions: in late spring (June) and autumn (October).

### 2.2. Putative Psa Isolation and Total DNA Extraction

Four kiwifruit leaves with typical symptoms of bacterial kiwifruit canker that include small, angular water-soaked areas, and chlorotic halos around necrotic foliar spots were collected from three individual kiwifruit plants from each orchard and saved in separate sterile bags. Sampled plants were marked, and their Global Positioning System (GPS) position was recorded. The collected samples were labelled, transported into the laboratory at 4 °C, and processed immediately.

Plant samples were processed to separately recover epiphytic and endophytic Psa strains. Leaves were swabbed with Phosphate Buffered Saline (PBS) containing 0.01% Tween 20 to remove biofilms and efficiently isolate the epiphytic bacterial communities. Swabs from the same plant were combined, shredded, and vortexed for 5 min in 9 mL of sterile 10mM PBS to prepare the epiphytic bacterial suspensions.

After recovering epiphytic bacteria, leaves were sterilized as previously described by Eevers and colleagues [40]. Briefly, leaves were washed for 3 min in sterile Milli-Q water, 1.5 min in ethanol 70%, 3 min in NaOCl 1%, 1.5 min in fresh ethanol 70%, and finally rinsed five times with sterile Milli-Q water. The last rinsing water was inoculated in King’s B medium (KB) and incubated at 25 °C for 72 h to confirm the efficiency of the process. Sterilized leaves were shredded in a blender (approximately 1 mL of sterile 10 mM PBS per g of plant samples) and the obtained macerate was passed through a sieve (sterile gauze cloth) to remove plant debris.

The obtained suspensions (epiphytic and endophytic) were serially diluted (up to 10^−6^) and aliquots of 100 µL were plated onto KB medium supplemented with cycloheximide (200 mg/L), cephalexin (200 mg/L) and boric acid (0.15%) (KBc) and incubated at 25 °C for 72 h and inspected for colony growth every 24 h.

From each 10^−1^ serial diluted plate, ten Psa-like isolates were selected and further purified twice on KB. On KBc, Psa-like colonies appear smooth, flat, with entire or slightly lobed margins, pearly whitish-yellowish in colour, and 4–5 mm wide after 4–5 days, showing a tiny white spot at the centre of the colony [41]. To increase diversity, colonies with different morphology were also selected from the remaining serial diluted plates and further purified by re-streaking on KB as often as necessary to obtain pure cultures. All strains were cryopreserved.

For molecular analysis, a bacterial suspension from single colonies of each isolate was prepared in lysis buffer (Tween 20 2% solution in NaOH 0.1 M) and total DNA was extracted by heat-denatured treatment. The total DNA concentration was normalized following nanodrop quantification.

### 2.3. Psa Molecular Identification and Typing

A duplex-Polymerase Chain Reaction (PCR) with the primers KN-F/KN-R and AvrDdpx-F/AvrDdpx-R, developed by Koh & Nou [42] and Gallelli et al. [43], yielding 492 and 226 bp amplicons, respectively, was used to identify Psa strains. BOX-PCR was performed as described by Louws et al. [31] to assess the genetic diversity of Psa strains. When no signal was obtained with the standard BOX-PCR conditions, several concentrations of primers and total DNA were tested. Psa reference strains CFBP7286 (biovar 3, isolated in 2008 in (Lazio Latina, Italy) [1], and CFBP7812 (*Pseudomonas syringae* pv. *actinidifoliorum* (previously known as biovar 4), isolated in 2010 in Nelson Motueka (New Zealand) [20], were included in each BOX-PCR analysis for comparison purposes. BioNumerics software was used for gel image processing and normalization. Clusters were formed by visual inspection based on the similarity and intensity of the fluorescence of each band observed in Psa profiles, namely the number and weight of the bands when compared with each other, with the reference strains, and with the molecular weight marker. The sequences of the primers used in this study are presented in Appendix A. PCR was conducted with a MycyclerTM Thermal cycler (Bio-rad). PCR products were separated by horizontal gel electrophoresis in a 2% agarose gel (*w*/*v*) in Tris-Acetate-EDTA (TAE) (1%) buffer and staining with ethidium bromide. To estimate the size of the amplicons, the molecular weight marker NZYDNA Ladder III (Nzytech, Lisbon, Portugal) was used. The DNA bands were visualized with Image analyser DocTm XR+ (Bio-Rad, Amadora, Portugal).

### 2.4. Molecular Characterization Psa Strains

Molecular tests were performed on the 30 representative Psa strains selected from the previously established clusters based on the BOX-PCR fingerprinting analysis. One representative strain was selected from each BOX-PCR profile with less than 10 strains if they were detected in one orchard; for BOX profiles with more than 10 strains, one strain was selected from each orchard. Psa reference strains CFBP7286 and CFBP7812 were included in each.

Four sets of primers were used to determine the biovar of each representative Psa strain according to the multiplex-PCR protocol described by Balestra et al. [44] (Appendix A). Reference strains with known biovar were included in the analysis, namely strain Psa CFBP4909 (biovar 1), Psa ICMP19071 (biovar 2), Psa CFBP7811 (biovar 3), and *Pseudomonas syringae* pv. *actinidifoliorum* CFBP7812 and CFBP8044 (both former Psa biovar 4, Psa b4L1) (Appendix A).

The presence of the genes coding for toxin coronatine (*cfL*) and phaseolotoxin (*argK*) was determined using the primers CFLF/CFLR [45] and ArgKF3/ArgKR [46], respectively (Appendix A).

The purity and yield of each amplicon were verified into 1.5% agarose gel in 1×Tris-Borate-EDTA (TBE) buffer. PCR products were visualised with Gel DocXR + BioradImager (Bio-Rad, Amadaora, Portugal).

### 2.5. Multi-Locus Sequencing Analysis(MLSA) of Psa Strains

MLSA was performed according to the protocol previously described [21,22,47] based on the partial sequence of four housekeeping genes: glyceraldehyde-3-phosphate dehydrogenase (*gapA*), citrate synthase (*gltA* or *cts*), DNA gyrase B (*gyrB*), and sigma factor 70 (*rpoD*), yielding 675 bp, 995 bp, 674 bp, and 812 bp amplicons, respectively (Appendix A). The purity and yield of each amplicon were verified into 1.5% agarose gel in 1× TBE buffer. PCR products were visualized with Gel DocXR + BioradImager (Bio-Rad, Amadora, Portugal), purified using NZYGelpure extraction kit (Nzytech, Lisbon, Portugal) and sequenced by Sanger’s platform, as a contracted service (STABVIDA, Caparica, Portugal). The quality of the sequences was manually checked using the Sequence Scanner software (Thermofisher Scientific, Porto Salvo, Portuga).

The four loci were concatenated in single multiple alignments with BioEdit 7.1.9 [48], following the alphabetic order of genes, as described by Abelleira et al. [22]. The concatenated data set was 2159 bp long (in the alignment: 1 to 643 bp *gapA*, 644 bp to 1106 bp *gltA*, 1107 bp to 1683 bp *gyrB*, and 1684 bp to 2159 bp *rpoD*). Sequences from several reference strains obtained from public databases were included in this phylogenetic analysis (Appendix A). The alignment was performed using the multiple alignment CLUSTAL software [49], included on MEGA X version 10.0 [50].

Neighbour-joining (NJ) phylogenetic tree was built with MEGA X version 10.0 [50] using the Jukes-Cantor distance methods from the concatenated alignment. Supports for the nodes were evaluated by bootstrapping with 1000 pseudoreplicates. The strain of *P. syringae* pv. *tomato* CFBP 2212 [51,52] was used to root the trees.

### 2.6. Statistical Analysis

The relationships between the relative abundance of Psa isolates (response variable) and the environmental variable’s orchard, location in the leaf (epiphytic and endophytic), and season (spring and autumn) were analyzed using a generalized linear model (GLM) regression with 95.0% confidence, where *p*-values below 0.05 were statistically significant. The confidence intervals and standard errors of each mean were also calculated. Multiple comparisons were performed with Fisher’s least significant difference (LSD) to determine which means were significantly different. All analyses were run on R 4.0.1 [53]. The characterization of Psa population diversity, within and between orchards, was supported by the measurement of alpha and beta diversity, as proposed by Whittaker [54]. Alpha diversity analysis [55] was individually performed for each orchard to determine the overall diversity and compare Psa population diversity between plants per sampling condition (considering epiphytic and endophytic isolates in two distinct seasons—spring and autumn). Margalef index (Dmg), Shannon index (H’), Pielou’s evenness index (J’), and Simpson diversity index (1-D) values [56] were calculated using the package ‘vegan’ for R software v. 2.4–6 [57].

The distribution of Psa populations over time, within and between orchards, and their distribution as epiphytic or endophytic populations were assessed through the calculation of indirect, linear response model principal component analyses (PCA)—inter-species correlation, based on Psa BOX profiles data using the software package CANOCO 5 [58].

### 2.7. Nucleotide Sequence Accession Numbers

The complete *gapA*, *gltA*, *gyrB*, and *rpoD* sequences from Psa strains determined in this study were deposited in the GenBank Database (accession numbers MN916589–MN916708).

## 3. Results and Discussion

### 3.1. Psa Isolation from Portuguese Kiwifruit Orchards

Four orchards of *Actinidia deliciosa* infected with *Pseudomonas syringae* pv. *actinidiae* (Psa) from different regions of continental Portugal were selected based on the location, age, degree of disease severity, and cultivar (Table 1). Samples for the detection of Psa were collected from the same plants twice in 2016, in spring and the following autumn. Moreover, the epiphytic and endophytic populations were separately analyzed.

Despite the use of a defined recognized strategy, including a selective medium for the isolation of Psa, more than 50% of isolates obtained in each orchard did not confirm to be Psa. A total of 970 putative Psa strains were isolated from the four studied kiwifruit orchards: 247 isolates from orchard A, 203 isolates from orchard B, 263 isolates from orchard C, and 257 isolates from orchard D. Of these, 478 were isolated in spring and 492 in autumn. We isolated 430 endophytic and 540 epiphytic strains (Appendix A). From the total isolates, 604 (62.3%) were confirmed and identified as Psa according to Gallelli et al. [43] (Appendix A).

Overall, a statistically significant relationship between the relative abundance of Psa isolates and the predictor variables orchard, season, and location on the leaf was determined by GLM regression (*F* = 6.15, *p*-value = 0). Moreover, the combination of the factor’s orchard/leaf location (*F* = 4.85, *p*-value = 0.007) and season/leaf location (*F* = 11.2, *p*-value = 0.0021) affected significantly the relative abundance of Psa isolates. On the contrary, no statistical support was observed for the factor’s orchard/season (*F* = 1.29, *p*-value = 0.29).

In terms of multiple comparisons, the relative abundance of Psa was significantly different (*F* = 3.08; *p*-value = 0.041) between orchards B (10.83 ± 4.01) and C (15.91 ± 4.02), and between orchards C (15.91 ± 4.02) and D (10.67 ± 4.03). Moreover, the distribution of Psa isolates varied significantly (*F* = 42.43; *p*-value = 0.00005) with the leaf location (endophytic17.13 ± 2.84 and epiphytic 8.04 ± 2.84). No significant differences were detected in the abundance of Psa isolates between seasons.

A sour major goal was to characterize the genetic structure of Psa populations and to advance the knowledge on the impact of seasons and location in the leaf in this diversity, the remaining analyses were carried out exclusively with the confirmed Psa isolates (Appendix A).

### 3.2. Diversity of Psa Populations in Portuguese Kiwifruit Orchards

The molecular typing of the recovered Psa isolates was performed with BOX-PCR fingerprinting analysis [31], as it has been widely used to assess the genetic diversity between Psa isolates and to determine the genetic relationship with Psa strains isolated worldwide [20,34,38,59]. In accordance, Psa reference strains and related pathovars displayed different BOX profiles when analyzed, confirming the discriminatory ability previously described for this methodology [20,34,38,59]. From the 604 Psa isolates from the four kiwifruit orchards, we obtain reproducible and robust BOX profiles out of 549 Psa isolates, despite several attempts (see Materials and Methods section).

Overall, the Psa population genetic diversity was determined based on the BOX-PCR profiles obtained from 133 isolates from orchard A, 112 isolates from orchard B, 178 isolates from orchard C, and 126 isolates from orchard D (Appendix A). This analysis generated 16 different BOX profiles, all distinct from those obtained for the reference strains, revealing the existence of a heterogeneous Psa population, with high genetic variability (Figure 1). This high genetic diversity was mainly owing to the high number of Psa profiles exclusively obtained for strains from orchard A and B, combined with populations common to more than one orchard (Figure 1, Appendix A).

The data also show the existence of mixed Psa populations in each orchard. For instance, twelve different profiles were identified in orchard A, eight in orchard B, five in orchard C, and four in orchard D (Figure 1). Moreover, the overall Psa diversity was remarkably distinct between kiwifruit orchards, as only three Psa profiles (P) were found in all orchards, namely, P5, P13, and P36. These three profiles were dominant in orchards C and D, corresponding to 94% of Psa isolates, while they were more evenly distributed in orchards A and B, corresponding to 46% and 71% of the total Psa isolates, respectively (Figure 1). Four other profiles were detected only in some orchards, namely, P9 obtained for strains from orchards A and B, P6 from orchards A and C, P7 in orchards B and C, and P33 in orchards B and D (Figure 1 and Appendix A).

A principal component analysis (PCA) was performed to establish a potential correlation between the orchards considering their Psa genetic profile diversity (Figure 2). With this analysis, it was possible to correlate the influence of each Psa population, in the diversity of each orchard.

In detail, Psa populations from orchards C and D were related and distinct from those found in the northern region orchards (Figure 2A). This configuration was mainly owing to the high abundance of P5 and P13 strains in theses orchards when compared with the lesser abundance in orchards A and B. Moreover, orchard A was completely distinct from the others given the presence of several isolates with unique profiles that contributed significantly to the increase in diversity, namely, P2, P3, P4, P8, P10, P23, and P24. Conversely, Psa populations in orchard B were significantly different from the others, despite sharing profiles with all of them. This distinctiveness was mainly due to the presence of strains with two exclusive profiles (P27 and P37), and the higher abundance values for profiles P7, P9, and P36 (Figure 2A; Appendix A).

Our observations were supported by alpha diversity indexes calculated for each orchard (Table 2A). According to Margalef and Shannon indexes, orchards from the northern region (A and B) presented higher Psa diversity than the orchards from the central region (C and D) (Table 2A). Furthermore, Psa diversity was more evenly distributed in the northern region, with Pielou’s evenness values of 0.8 and 0.7 for orchard A and B, respectively. The orchards from the central region, on the other hand, present lower Psa profile diversity, associated with the higher abundance of isolates with dominant profiles, evidenced by the Simpson index with values of 0.40 and 0.50 for orchard C and D, respectively (Table 2A).

### 3.3. The Diversity of Psa Populations Varies between Seasons in Each Orchard

To determine if the Psa population structure varied between seasons, Psa isolates were recovered in consecutive spring and autumn. An additional PCA analysis was performed considering the period of sampling to assess the impact of the season in the Psa diversity orchards (Figure 2B). From this analysis, it was clear that the abiotic conditions affected Psa diversity in all orchards, inducing similar changes because we could identify two distinct patterns. While in spring, Psa populations were quite distinct, presenting strains with unique profiles that split them in the PCA analysis, in autumn, they were clustered together mainly by the presence of common and dominate populations with similar profiles, namely P5, with a concomitant decrease in overall Psa diversity.

This trend was observed for all orchards (Figure 2B) and supported by alpha diversity indexes because, in spring, higher richness (Marglef diversity index) was observed, isolates were more evenly distributed (Pielou’s diversity index), and the alternative dominance (Simpson diversity index) was higher (Table 2). Nevertheless, this pattern was particularly explicit in orchard A inferred from twelve profiles (the highest observed diversity among spring isolates), in contrast to only three profiles recovered from autumn isolates. This drastic reduction in diversity was owing to a clear predominance of isolates with P5 among autumn isolates (Figure 2B). Once again, this trend was most evident in orchard A, as P5 corresponded to 1.5% of the spring isolates, reaching 76% of the autumn isolates. This dominance was observed in all orchards, reaching alternative Simpson diversity values of 0 in orchards A and D (Table 2B). Indeed, P5 represented 39%, 61%, and 48% of the spring isolates from orchards B, C, and D, respectively; while in autumn, it reached 85%, 92%, and 100% of the isolates, respectively. This decrease in Psa variability could be related to variations in the abiotic conditions affecting the orchards between spring and autumn, namely higher temperatures and less humidity (summer conditions), suggesting that the prevalence of P5 could be related to its fitness to overcome such conditions. Only a few profiles were recovered in both seasons in the same orchard, namely, P6, P7, P13, P23, and P27 (Figure 2B).

On the contrary, P36, common to all orchards, was only detected during spring, suggesting its inability to cope with the summer conditions known to affect Psa proliferation [24,26]. Indeed, this trend was observed for several other profiles, as they were only recovered in a specific season. Namely, P2, P3, P4, P8, P9, P10, P24, and P33 were only detected in spring, while P37 was exclusively identified in autumn. These results suggest the existence of differences between the various Psa profiles in their capacity to persistence/thrive in the plant that could be related to the changes in the abiotic conditions observed between seasons. Recently, Straub and colleagues [60] described not only the influence of the infection status and cultivar but also the potential impact of *P. syringae* commensal microbes in the Psa population structure and infection process. Indeed, 40% of our isolates were identified as putative *Pseudomonas* sp., supporting this ecological perspective.

### 3.4. The Diversity of Psa Populations Varies in the Phyllosphere

In addition to the structural changes that were previously described for the Psa population over time, the diversity also varied with the location in the phyllosphere. The methodological approach proved to be fitted to study the Psa populations from these two distinct niches because we did not recover any isolate from the last rinsing water, confirming the sterility of the leaves’ surface (see Materials and Methods section).

To determine the distribution of Psa populations in the phyllosphere, a PCA analysis was performed based on the profiles obtained from the Psa isolates from the epiphytic and endophytic niches (Figure 2C). Overall, differences between the epiphytic and endophyte Psa population structure were observed in samples collected simultaneously from the same orchard, because some populations were able to persist epiphytically and endophytically, while others were exclusively found in one of the niches. Psa populations from orchards C and D were grouped mainly by the dominance of strains with P5, P13, and P36. On the other hand, the epiphytic and endophytic populations from orchards A and B were rather distinct owing to the presence of P4, P6, P8, P13, P24, P27, and P33 exclusively detected in one of the niches (Figure 2C).

The majority of the Psa populations were able to persist epiphytically and endophytically. This wide distribution in plant leaves was predominant in spring, with several populations isolated from both niches, namely, strains with P2, P3, P9, P13, and P36 in orchard A; P5, P7, P9, and P36 in orchard B; P5, P13, and P36 in orchard C; and P5, P13, P33, and P36 in orchard D. On the contrary, only two populations were able to persist both epiphytically and endophytically in autumn, namely P5 in all orchards and P7 in orchard C. Curiously, in orchard A, isolates with P5 were not detected among endophytic spring isolates and accounted for just 2.8% of the epiphytic spring isolates, but became the only population detected in autumn from epiphytic isolates. This limited distribution in spring could be related to the cultivar present in orchard A, as no other obvious difference could explain this limited distribution [60].

However, some BOX profiles were exclusively detected in one of the niches. Regarding Psa profiles with a limited distribution in kiwifruit leaves, P4, P8, P24, and P27 were exclusively detected in epiphytic isolates, while P6, P10, and P37 were exclusively found in endophytic isolates (Appendix A). The majority were spring isolates from orchard A, namely, those with P4, P6, P8, P10, and P24, while strains with profiles 6 and 27 were shared between orchards A and C, and between orchards A and B, respectively.

Moreover, the epiphytic and endophytic Psa population structure varied between seasons. This trend was observed for all orchards and supported by alpha diversity indexes consistently lower in autumn for all orchards (Table 2B). According to Margalef and Shannon indexes, during spring in orchards A and B, the Psa epiphytic population was more diverse and evenly distributed when compared with the endophytic population (Table 2B). On the contrary, the endophytic Psa population was more diverse in orchards C and D during spring, but with an overall lower diversity compared with the orchards from the north region (Table 2B). A profound change in Psa diversity was observed in autumn, namely in orchards A and D, and to a lesser extent in orchard C. This was associated with the higher abundance of isolates with dominant profiles evidenced by a lower Simpson index for all orchards (Table 2B). The decrease in Psa diversity in autumn was most noticeable among epiphytic isolates from orchard A and among epiphytic and endophytic isolates from orchard D, as only one population (P5) was detected. This was supported by Simpson’s indexes of 0.0 (Table 2B).

### 3.5. Psa Population Structure in Individual Plants

The Psa population structure was also determined for each of the three sampled plants from each of the studied orchards. Only four of the sixteen Psa profiles were exclusively found in isolates from a single plant; namely, P4, P8, P24, and P37. All the remaining profiles were detected in isolates from at least two plants (Appendix A). This was particularly evident in plants from orchard A, as no strains with the same profile were found in all plants. Indeed, most of the profiles were found in isolates associated to a single plant, namely, P3, P4, P8, P9, P23, P24, and P36, while profiles P2, P5, P6, P10, and P13 were shared by isolates from at least two plants (Appendix A).

Fewer differences were observed among plants from orchards B, C, and D because they shared isolates with several profiles; nevertheless, some strains with unique profiles were also isolated from single plants. That is, in orchard B, five profiles were found in isolates from at least two plants, specifically P5, P7, P9, P13, and P36, while three profiles were restricted to strains from a single plant, namely, P27, P33, and P37 isolated from plant 2, 3, and 1, respectively. Isolates from orchard D presented the lowest number of Psa profiles, with a corresponding lower richness index, comprising isolates with only four profiles; two shared by at least two plants, P5 and P13; and two others found in isolates from a single plant, namely P33 and P37, recovered from plant 1 and 2, respectively. The most homogeneous orchard was orchard C because all Psa populations were shared by at least two plants.

Moreover, we also analysed the epiphytic and endophytic populations from each plant in spring and autumn. In general, there were common profiles between epiphytic and endophytic isolates from each plant in each season. This was clear in spring for most of the plants, with several shared profiles between niches. On the contrary, in autumn, only P5 strains were common between epiphytic and endophytic isolates.

Nevertheless, some strains with unique profiles were only detected in one of the niches. This was particularly evident in orchard A as several profiles were only found to be associated to epiphytic isolates (P4, P8, P23, and P24), while others were only found in endophytic strains (P6 and P10). These results evidence the co-existence of Psa populations, although the distribution of some profiles may indicate the specificity of some strains for particular niches. The results also showed that some Psa populations changed with time, while others were persistently recovered.

In sum, Psa populations varied with seasons and leaf location in the same plant. Higher Psa population’s diversity was found among spring isolates when compared with those isolated in autumn.

### 3.6. Molecular Characterization and Phylogenetic Analysis of Selected Psa Strains

Thirty Psa strains were selected from the BOX fingerprinting analysis to be further characterized based on additional molecular tests. We concluded that, despite the heterogeneity reported in this study, all Psa strains belonged to biovar 3, according to Balestra et al. [44] protocol. Moreover, this result was supported by the lack of phaseolotoxin and coronatine genes, commonly used to distinguish Psa biovars [14,16].

Moreover, the Portuguese Psa population structure was inferred from the concatenated partial sequences of four housekeeping genes *gapA*, *gltA*, *gyrB*, and *rpoD* [46] and compared with selected reference strains (Appendix A). Three discrete groups were obtained with this analysis, corresponding to distinct *P. syringae* pathovars, namely Psa (Group A), *P. syringae* pv. *actinidifoliorum* (Group B), and *P. syringae* pv. *tomato* (Group C) (Figure 3). Strains classified as Psa were grouped into two main clusters within Group A: Cluster I, comprising the majority of our isolates and the reference strains from biovar 1, 3, and 6. As previously reported, this MLSA-based analysis is not able to discriminate Psa biovars 3 and 6, as they formed a single cluster [12,20]. These results agree with the molecular analysis described above. Cluster II included the reference strains from biovar 5 and 2, each corresponding to a discrete group, as previously described [11,61].

The current perspective of Psa3 evolution encloses several stages; originally, the diversification was restricted to natural environments in China through the acquisition of exogenous DNA with the selection of organisms with increased fitness [18,23]. The admission of distinct and independent transmission events supported by genomes comparison marked the following stage with the introduction of the pandemic lineage into New Zealand, Chile, Europe, Korea, and Japan [18,19,32,62]. In Europe, and according to recent studies on comparative genomics, Psa3 was clonally spread without foreign contribution in an ecological niche lacking competition with diversification thought rearrangement of self-genetic elements without any gene gain [19]. The results presented here support the diversification of Psa3 during the relatively short time since the initial introduction, identifying clones more fitted to specific abiotic conditions. These differences could result from the rearrangement of self-genetic elements, as previously proposed [19]. Nevertheless, McCann et al. [23] reported that approximately 10% of the pandemic Psa3 genome shows evidence of homologous recombination marked by gene conversion envision other possible mechanisms responsible for Psa diversification. The co-existence of Psa3 strains with other *Pseudomonas* sp. in kiwifruit leaves as described herein may constitute a melting pot favouring recombination events driving the selection of more fitted strains. Indeed, Psa biovar 3 phylogeny is consistent with diversification from a single clone, with a dynamic genome revealing evidence of gain and loss via multiple genetic routes [18,23,63]. This supports the current view that Psa emerged has a crop pathogen from recombining with not only pathogen populations specific to the host plant [23,64,65], but also *P. syringae* environmental populations that colonize multiple hosts and are widely distributed among both plant and non-plant habitats, that is, soil, irrigation water, and decaying plant material [19,63,64,65]. Indeed, the emergence of the pandemic lineage did not displace more ancestral strains as both pandemic and divergent Psa strains were isolated in China [18,23,63].

This is relevant information that should be taken into consideration when assessing the long-term strategies to be adopted for the management and control of the kiwifruit bacterial canker caused by Psa. Further genomic studies will unravel the origin of the genetic diversity found in the Portuguese Psa strains.

## 4. Conclusions

In conclusion, our results demonstrate a previously undescribed high genetic variability between Psa isolates with several co-existing populations. Moreover, obvious changes in the population’s structure occurred between leaf niches and seasons, favouring the dominance of some Psa strains in autumn. These findings suggest the existence of differences between the various Psa populations in their capacity to persist/thrive in the plant. Multiple factors can contribute to the observed diversity, including epidemic dynamics or disease management. The biovar introduced in the territory influences the success of the mitigation measures that can be enhanced using tolerant cultivars, containment measures aimed at reducing the inoculum, and controlling whenever possible the conditions favourable to dispersion [3]. Moreover, changes in abiotic conditions are known to influence the presence and multiplication of Psa in kiwifruit plants [8,25] and were described herein. Furthermore, the microbial community present in kiwifruit plants is another factor potentially influencing host-pathogen relation [25]. We should consider these as major factors as abiotic environmental filtering, spatial (niche-based) processes, and competition (microbiota) is expected to shape the evolution of Psa. In our context, the diverse genetic pool could be shaped by these factors selecting Psa clones according to their fitness. This trend is supported by the observed decrease in Psa diversity in autumn with the concomitant selection of dominant strains. This new perspective is important for a more comprehensive understanding of kiwifruit bacterial canker disease occurrence and Psa evolution. It is also relevant when adopting strategies for management of epidemics.

## Figures and Tables

**Figure 1 microorganisms-08-00931-f001:**
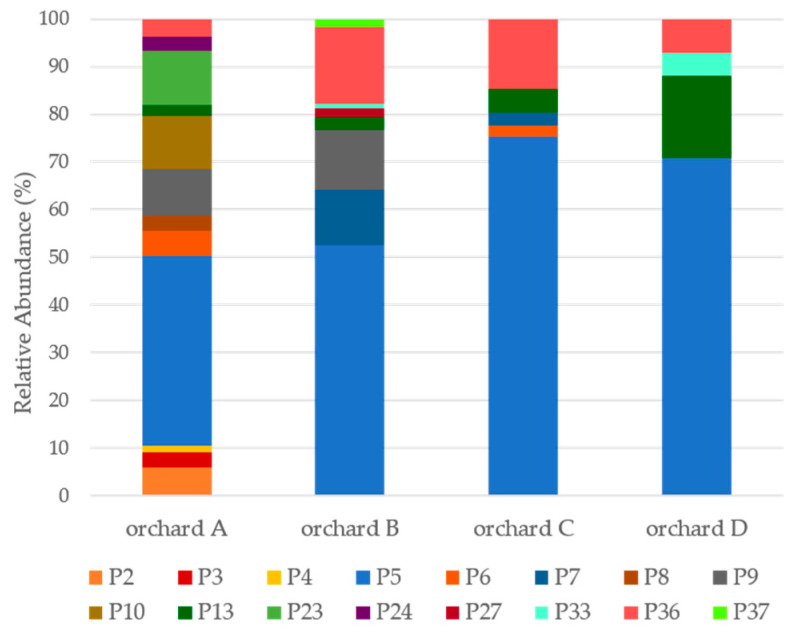
Psa population structure in orchard A to D.

**Figure 2 microorganisms-08-00931-f002:**
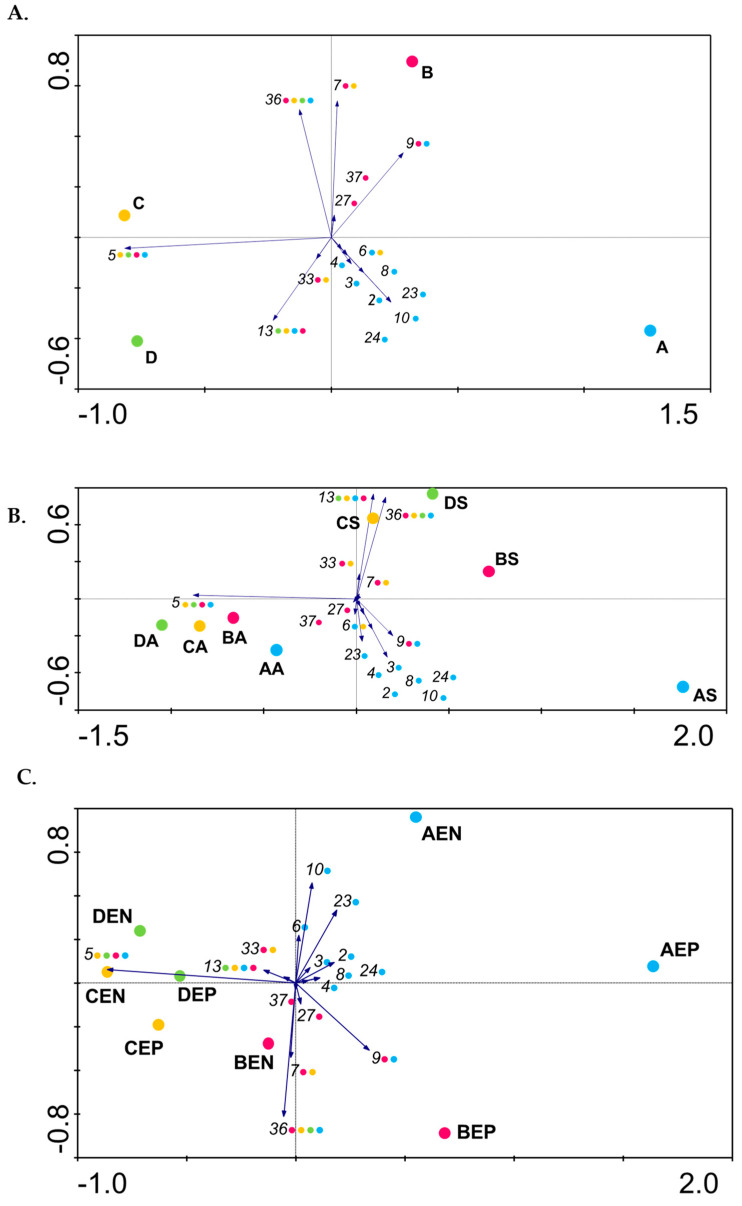
Principal Coordinates Analysis (PCoA) plots generated from Psa profiles identified in the four studied orchards using CANOCO 5 [58]. PCoA plots were constructed to (**A**) ascertain the distribution of Psa populations within and between orchards; (**B**) the distribution of Psa profiles in spring and autumn; and (**C**) the distribution of Psa profiles as epiphytic or endophytic profiles in the different orchards. Numbers correspond to Psa profiles and arrows identify the weight that each profile had on the diversity relationship between orchards. Blue, profiles isolated from orchard A (A); pink, from orchard B (B); yellow, from orchard C (C); green, from orchard D (D). A, autumn; S, spring; EN, endophytic; EP, epiphytic.

**Figure 3 microorganisms-08-00931-f003:**
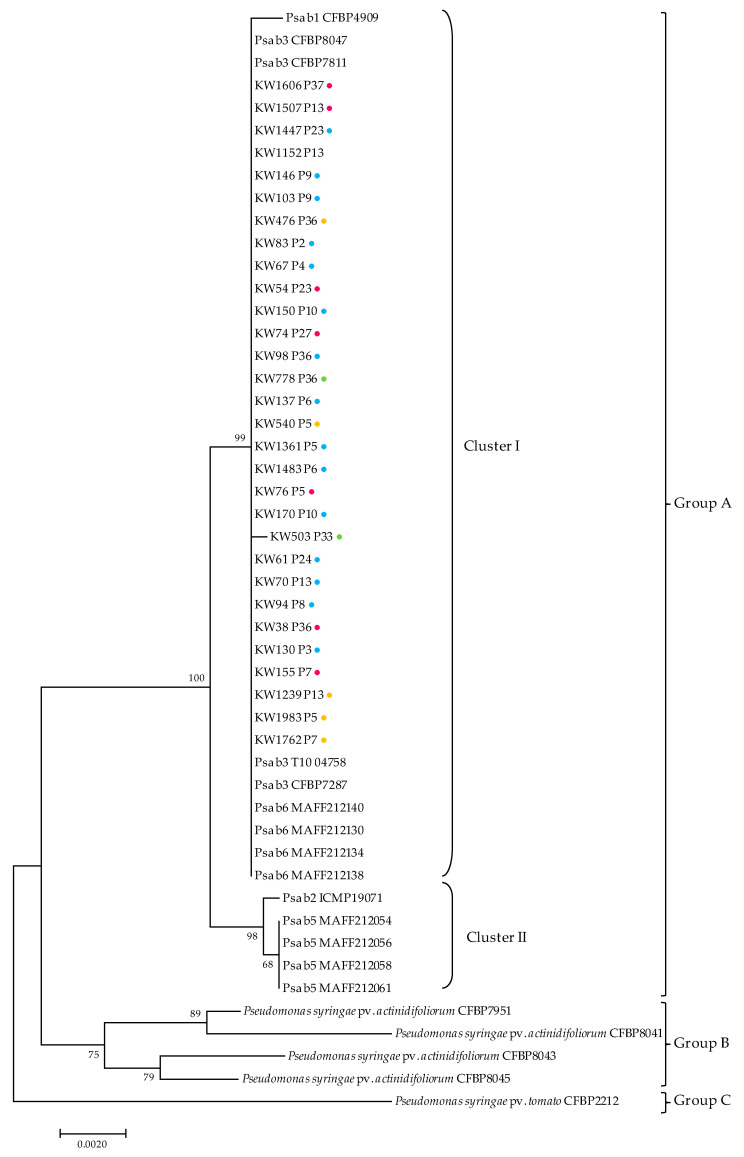
Maximum likelihood tree inferred from the concatenated partial sequences of four housekeeping genes (*gapA*, *gltA*, *gyrB*, and *rpoD*) from selected Psa strains (Appendix A). *P. syringae* pv. *tomato* was used as outgroup. Labels P1 to P36 indicate the Psa BOX profile; b, biovar; bootstrap support values (1000 replicates) are indicated at each node. Scale bar, 0.02 inferred nucleotide substitutions per nucleotide position. Blue dot, profiles isolated from orchard A; pink, from orchard B; yellow, from orchard C; green, from orchard D.

**Table 1 microorganisms-08-00931-t001:** Characteristics and climatic conditions of the studied Portuguese kiwifruit orchards.

Orchard	Localization	Cultivar	Age (Years)	Psa Detection (Year)	Psa Disease Severity Degree ^1^	Average Annual Temperature (T°) ^2^	Annual Cumulative Rainfall (mm) ^3^	Number of Cold Hours (h) ^4^
A	North	Bo.Erika^®^	7	2010	1	13.5	1800	478
B	North	Hayward	5	2015	2	12.5	1800	1031
C	Centre	Hayward	4	2015	3	14.5	1100	541
D	Centre	Hayward	30	2016	2	14.5	1100	440

^1^ Adapted from a symptomatology scale (0—asymptomatic plants to 4—completely dry plants) used in pathogenicity assays by Cunty et al., [21]; ^2^ average annual temperature; ^3^ annual cumulative rainfall: normal of 1961/90; ^4^ number of cold hours: total number of hours of T °C below 7.2 °C between 01/10/2015 and 30/04/2016.

**Table 2 microorganisms-08-00931-t002:** Alpha diversity indexes from orchards (**A**) season and leaf location (**B**), obtained from the dataset matrix of fingerprinting generated by BOX-PCR of Psa strains isolated from the four kiwifruit orchards analyzed in this study.

A.						
**Orchard**	**Index**					
	**Dmg ^1^**	**H** **’ ^2^**	**J** **’ ^3^**	**1-D ^4^**		
A	2.1	2.0	0.8	0.8		
B	1.5	1.4	0.7	0.7		
C	0.8	0.8	0.5	0.4		
D	0.6	0.9	0.6	0.5		
B.						
**Orchard**	**Season**	**Leaf Location**	**Index**			
			**Dmg ^1^**	**H’ ^2^**	**J’ ^3^**	**1-D ^4^**
A	spring	EP ^5^	2.4	2.1	0.9	0.9
		EN ^6^	1.5	1.5	0.8	0.7
	autumn	EP	0	0	0	0
		EN	0.5	0.8	0.7	0.4
B	spring	EP	1.3	1.5	0.8	0.7
		EN	1.1	1.4	0.9	0.7
	autumn	EP	1.1	0.9	0.8	0.5
		EN	0.6	0.4	0.4	0.2
C	spring	EP	0.6	0.8	0.8	0.5
		EN	0.7	1	0.7	0.6
	autumn	EP	0.3	0.4	0.6	0.3
		EN	0.5	0.2	0.2	0.1
D	spring	EP	0.6	1	0.7	0.5
		EN	0.7	1.2	0.9	0.7
	autumn	EP	0	0	0	0
		EN	0	0	0	0

^1^ Dmg, Margalef index; ^2^ H’, Shannon index; ^3^ J’, Pielou’s evenness index; ^4^ 1-D, Simpson diversity index; ^5^ EP, epiphytic isolates; ^6^ EN, endophytic isolates.

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
