# Peer review of "Genetic Diversity of Pseudomonas syringae pv. actinidiae: Seasonal and Spatial Population Dynamics"

_microorganisms, 2020, doi:10.3390/microorganisms8060931_

Round 1
Reviewer 1 Report
A very high number of isolates (549) has been characterized by BOX-PCR to study the genetic variability of Psa Portuguese population. These revealed 16 BOX profiles different from that of the reference strains used, revealing a high genetic diversity. Previous studies used for the investigation of genetic variablity of Psa population used a more limited number of strains. So this paper give a higher overview of the genetic diversity of Psa, and although restricted to Portuguese strains, give the indication of a higher variability than that known up to now within Psa population. Finally, 30 Psa selected strains were further investigated by MLSA analysis confirming previous molecular analyses.
The novelty of this paper is the high genetic variability associated to Psa bacterial population possibly related to seasonality and leaf location (epiphytic and endophytic populations); a lot of data has been recovered that needs to be better fixed in the text. Some evidences are not clearly supported from the results of this paper, for example the sentence in the abstract “possibly explained by the coexistence with other Pseudomonas sp. and relatives in kiwifruit leaves” (line 20-21), or in the conclusion “the susceptibility of the plant host, namely the genetic characteristics of the plant (cultivar) and physiological status, are also decisive factors influencing the colonization” (lines 494-495), and in the results and discussion: “we could also hypothesize that our results could be explained as a direct consequence of changes in the abiotic conditions and/or by alterations in the community influencing the Psa population genetic structure” (lines 334-336), but also " orchards practices that varied beteen spring and autumn that could affect the Psa populaton structure, the community..." (lines 361-362). These latter aspects make this work somewhat speculative in its current form.
Furthermore, it is very difficult for the reader to follow the results and discussion in relation to Figure 1. A re-stiling of the paper is suggested, possibly referring to Figure 2 (more intuitive than Figure 1) and to the statistical data of Table 1. Throughout the text, the results refer to Fig. 1 which makes it very difficult to follow the contents, I suggest to relate the data and the consequent statements with Figure 2 (see comment 4 below) and Table 1. Table 1B is never mentioned in the results and in the discussion: it correlates the data of the BOX fingerprinting (therefore the Psa variability) with the orchards, the seasons, the epiphytic or endophytic location and therefore statistically supports some evidence in the text.
These latter evidences need to be better substantiated, some sentences reported at the end of the paragraphs (in particular 3.3,3.4, 3.5) seems to be a bit speculative (see point 2).
Major suggestions
-
The results of Psa isolation form Portuguese kiwifruit orchard (paragraph 3.1) needs to be statistically analysed to support the statements from lines 245-256.
-
The conclusions reported at the end of paragraph 3.3 (lines 334-336: “changes in abiotic conditions and alteration in the bacterial community”) and of paragraph 3.4 (lines 360-363: “consequence of changes in abiotic conditions combined with several implemented orchards practices”) seem speculative, no real evidences are produced that can correlates the differences in the bacterial population with abiotic condition/alterations in the bacterial community/physiological status of the plant or agronomic practices. These consideration reported at the end of each paragraph are not supported from robust data.
- It can also be interesting to better clarify the role of some profiles that showed a clear differences with respect others, and seem to be prevalent (e.g P5 which is highly represented in all season with the exception of orchard A in spring; or P36 which is high in spring and lower in autumn).
- The Figure 2 and table 1 are much more reliable than Fig. 1 in showing differences/significance of Psa diversity. Since figures 1 and 2 have been elaborated with the same data (correct?) I suggest to rewrite the results and the discussion on the basis of figure 2 (and tab. 1) which should provide the same scientific evidences of fig. 1, but are more intuitive and reliable. Otherwise it would be better to report only Fig 1A which gives a generic representation of the genetic variability of the Psa populations in the various orchards, leaving to only fig. 2 (and tab. 1) the role of highlighting the influence of the different parameters (seasons, plants, leaf location) on the genetic variability.
Minor suggestions
Material and methods
Lines 134-135
Four leaves were collected from each plant?
Result and discussion
Paragraph 3.1
Lines 245-246
This sentence can be referred to the epiphytic population, as there are no differences in the endophtic population between spring and autumn. However to support this statement a statistical analysis of the results of Fig S2 would be necessary.
Lines 246-247
Modify this statement based on the results of a statistical analysis: Psa isolates not were ALWAYS in high number in endophytic sample with respect epiphytic (see orchards A and B in spring). A statistical analysis is necessary to confirm these statments.
Lines 249-250
Also this sentence needs of a support of statistical analyses; is difficoult for a reader to follow these data in Figure S2, because is arduous to assign a score on the basis of the graph; however showing in details the values of orchards A and C the relative abundance of Psa isolates in A doesn’t seems higher than in C; I suggest to delete this sentence.
Line 252
“a significant reduction of the inoculum” the inoculum is not reduced if you find a high abundance of Psa in orchard A…
Lines 253-256
The results are difficoult to be verified with this kind of graph, very compilcated for the reader.
I suggest a statistical analysis of the results of Figure S2, reporting the standart error and the letter of significance on each column of results. This will allow to verify if the statements reported by lines 245-256 are reliable, otherwise it becomes difficult to accept the conclusions on the abundance of Psa in the 4 orchards and the relatives considerations.
Paragraph 3.2
Lines 310-311
“alternative dominance index” refer specifically to the index above reported to make more clear this sentence for the reader
Line 307/315/368
The adjective “notorious”…better to use clear/ evident?
Paragraph 3.5
Lines 383-384
“there were common profiles….from each plant, in each season”; an exception seems occur in plant 3, in spring in orchard A, so not on each plant/season there were common profiles. So this suggestion can be corrected indicating that was in the majority of the cases.
Paragraphs 3.2, 3.3, 3.4
The hypothesis that the results can related to abiotic conditions, alteration of the microbial community, implementation of agricoltural practices, the physiological status of the plant, season and leaf location seem, on the basis of the results of these paraghraphs, a bit speculative.
Paragraph 3.6
Line 429
Indicate which profiles it refers to, this helps the reader
Paragraph 3.6
Lines 447-448
Cluster II doesn’t exist. P. syringae pv. tomato belong to group C.
Figure and tables (supplementary and not)
Fig. S1 can be removed, the region and location (northern, central…) can be described in MM
Fig S2: column in black and white. A statistically analyses of these data is suggested.
Table 1B. footnotes 5 and 6 of acronym EP and EN (although easy to understand) are not reported
Conclusions
Lines 494-495
“the susceptibility of the plant host, namely the genetic characteristics of the plant (cultivar) and physiological status, are also decisive factors influencing the colonization” this sentence is not supported by the evidences of this work a citation of other works is nedeed.
References
It seem’s to me that the reported reference Gallelli et al., 2011 is not correct, probably you have to replace it with the following which refers to the development of the diagnostic method:
Gallelli A., L’Aurora A. and Loreti S., 2011. Gene sequence analysis for the molecular detection of Pseudomonas syringae pv. actinidiae: developing diagnostic protocols. Journal of Plant Pathology 93 (2): 425-435 , in which is reported method development.
Reviewer 2 Report
The manuscript Genetic diversity of Pseudomonas syringae pv. actinidiae: seasonal and spatial population dynamics deals witha an interesting approach to evaluate Psa diversity in different orchard with a culturable approach linked to the molecular characterization
The amount of work done is really relevant since more than 400 hundreds strains were included and indeed the results deserve interest and improve the knowledge on the epidemiology of the bacterium.
Both Material and methods and results are pretty clear exposed and the whole scientifically sounds. The results obtained with the population study (alpha-diversity) have very few references for discussion so if the authors feals they addressed or achieved very new data I encourage them to state it.
below some minor concerns.
L 160 subtitle proposal: Psa molecular identification and typing
L 166-67 ‘The obtained profiles were used 166 to cluster Psa isolates into homogenous groups and 30 representative strains were selected to be used 167 in the following analysis’. This could be omitted since explained thereafter.
L176 subtitle: There is no need to specify ‘representative’ in each title it could mentioned in the text once even in only in the results. Since the subtitle is similar to the one above it is better to propose something different e.g. Multiplex PCR for Psa biovar definition (if applicable)
L194 subtitle: remove representative
L213 I suggest you to combine in only one pargraph statistics
L 250 It is not clear what is the degree of the disease is it the incidence or the severity? Lower or hogher severity or incidence
L 273 instead of ‘found in orchard’ obtained for strains from
L 364 Psa population structure in ‘each’ plant /could be ‘inidividual’
Table 2 I'm wondering if this table add information. Is it possible to say that all the strains regardless the Orchard/plant and BOX profile all belong to the same MLSA cluster, gave positive amplification with………
Figure 3 legend: What do the coloured circles stand for?
Reviewer 3 Report
This paper presents the results of a study on the evolution of Pseudomonas syringae pv. actinidiae (Psa) in Portugal. The authors selected some symptomatic leaves in four orchards located in Portugal from which they isolated strains which had the characteristics of Psa (colony morphology and PCR). Strains were then classified according to their BOX PCR pattern.
This papers suggests some very interesting link between BOX PCR patterns and ability of some Psa strains to survive over time in the orchard (patterns found in spring samples versus those found in autumn samples) and their ability to cause infection (patterns linked with epiphytic populations versus those linked with endophyte populations). Those are big claims which are not completely supported by the evidence presented in the paper. The entire paper rests on the different BOX-PCR patterns observed among several Psa isolates, yet no electrophoretic pattern is presented and the differences between the 16 different patterns found by the authors are not presented either.
The MLSA analysis does not bring any new information (all the strains isolated in this study are similar to each other and to other strains of biovar 3). This part of the paper should be removed including Table 2 and Figure 3.
More detailed comments are listed below:
The Introduction could probably be more focused on the work presented in the paper: evolution of Psa.
In the Materials and methods section we need more details. For example L 121we need to know whether Erica is an Actinidia chinensis or a A. deliciosa cultivar. If table S1 was in the text of the paper then most of the section 2.1 could be deleted
L134 the authors should describe the symptoms on the leaves they sampled.
Similarly L 153 what is considered typical Psa-like colony morphology needs to be described in a few words.
L166 The strains CFBP 7286 and CFBP 7812 need to be described (biovar, year of isolation and country of origin). They are not listed in Table S3.
There is no description of any PCR protocol; did the authors always follow exactly the published PCR programmes? If this is the case then this should be mentioned.
BOX PCR can be difficult to interpret and results are not always reproducible; some of the minor bands might not be consistently detected. How many times each sample was analysed? Why is there no pictures of the profiles?
I would have preferred if the results were in a distinct section from the Discussion.
The message in Figure S2 would be easier to grasp if Psa was at the bottom, the information could be presented in a table.
L263 The authors could mention BOX PCR was used to distinguished Psa from P. syringae pv. actinidifoliorum (ref 20).
L266 How do the authors define reproducible and robust BOX profiles? How many technical reps were performed per strain?
L270 I wish we would have a pictures of the different profiles and a description of the differences between the profiles. Was it one band missing, was it a difference in the size or intensity of one or several bands? Could some profiles be clustered due to small differences while others were very different?
L318 not sure that it is necessary to abbreviate Spring Isolates to SI, since SI is mentioned only three times in the manuscript. Same for AI and Autumn isolates.
The authors suggest that some BOX PCR profiles are linked with bacteria which survive only epiphytically (P4, P8, P24 and P27), it would be very useful if the authors could inoculate strains from those profiles to determine whether they are capable of infection. Also why some profiles were found only associated with endophytic populations? Those bacteria must have been epiphytic at a time or another, so why they were not recovered as epiphytes?
How do we reconcile data given on L 356 ‘However, some BOX profiles were exclusively detected in one of the niches. Regarding Psa profiles with a limited distribution in kiwifruit leaves, P4, P8, P24 and P27 were exclusively detected in epiphytic isolates’; with those given on L389 ‘This was particularly evident in orchard A since several profiles were only found associated to epiphytic isolates (P4, P5, P8, P13, P23 and P24), while others were only found in endophytic strains (P10 and P13’ . From figure 1 I believe P5 and P 13 are found epiphytically and endophytically.
Round 2
Reviewer 1 Report
The authors implemented the manuscript taking into account the suggestions given. The manuscript in its current form has a clearer organization and the simplification in some parts allows a more fluid reading. Finally, having eliminated the speculative parts of the discussion, the conclusions are better supported by results.